# The Role of Supporting Cell Populations in Satellite Cell Mediated Muscle Repair

**DOI:** 10.3390/cells12151968

**Published:** 2023-07-30

**Authors:** Amanda L. Johnson, Michael Kamal, Gianni Parise

**Affiliations:** Department of Kinesiology, McMaster University, Hamilton, ON L8S 4L8, Canada

**Keywords:** satellite cell, skeletal muscle, regeneration, supporting cells, myogenesis, microenvironment, inflammation, vasculature, fibrotic cells

## Abstract

Skeletal muscle has a high capacity to repair and remodel in response to damage, largely through the action of resident muscle stem cells, termed satellite cells. Satellite cells are required for the proper repair of skeletal muscle through a process known as myogenesis. Recent investigations have observed relationships between satellite cells and other cell types and structures within the muscle microenvironment. These findings suggest that the crosstalk between inflammatory cells, fibrogenic cells, bone-marrow-derived cells, satellite cells, and the vasculature is essential for the restoration of muscle homeostasis. This review will discuss the influence of the cells and structures within the muscle microenvironment on satellite cell function and muscle repair.

## 1. Introduction

Skeletal muscle is a flexible organ that is capable of repair and remodelling in response to damage. It is composed of post-mitotic myofibres and, thus, cannot proliferate or divide. Instead, skeletal muscle relies on the local stem cell populations for support. It is generally agreed that the primary stem cell population that contributes myonuclei is satellite cells. These cells reside between the basal lamina and sarcolemma of myofibres in a state of dormancy or quiescence [1]. Upon activation, satellite cells can enter the myogenic program and proliferate, with the goal of donating myonuclei to a damaged myofibre or replenishing the pool of satellite cells [1,2].

Traditionally, muscle repair and regeneration are discussed within the context of satellite cell activity. While satellite cells are essential to the process, skeletal muscle repair is multifaceted and involves the interactions between several cell types and structures [3,4]. These components tend to reside within the interstitial space between fibres and generally act to support muscle maintenance as well as regeneration. Muscle connective tissue is responsible for maintaining structural integrity and the organized nature of skeletal muscle [5]. The connective tissue has three main components: fibroblasts, fibro-adipogenic progenitors (FAPs), and extracellular matrix (ECM) [6,7]. Fibroblasts are interspersed within the ECM and act as the main producers of ECM components, such as collagen and fibronectin [6,8,9,10]. FAPs are mesenchymal progenitors that are quiescent in homeostatic conditions and can differentiate into adipocytes and fibroblasts in response to stress [11,12,13,14]. The fate of FAPs is dictated by the stressor and the muscle environment. For example, FAPs are highly implicated in the elevated muscle fibrosis present in *mdx* mice (i.e., mouse models of muscular dystrophy) [9,15,16]. However, FAPs transplanted into obese mice underwent adipogenic differentiation, demonstrating that their differentiation potential is highly influenced by their environment [9,16,17].

Skeletal muscle is a highly vascularized tissue that is dependent on blood vessels to supply nutrients and oxygen as well as dispose of carbon dioxide and metabolic by-products. Capillaries, which run parallel to muscle fibres, are composed of a wall of endothelial cells surrounded by a basal lamina and pericytes. Pericytes, which are vascular supportive cells found in the basement membrane of microvessels, closely encircle endothelial cells and form connections with adjacent vessels [18,19,20]. These cells provide structural and paracrine support to blood vessels, regulate the permeability of capillaries, and modulate the constriction of microvessels [18,19,20,21,22]. Pericytes are considered multipotent mesenchymal progenitors as they can differentiate into multiple cell types depending on their environment [20,23]. The circulatory system is also responsible for the trafficking of immune cells to the myofibres. Neutrophils, macrophages, granulocytes, such as mast cells, and T-lymphocytes are involved in immune surveillance and have been linked to the inflammatory response within skeletal muscle [24,25]. There is a growing body of literature investigating the various cells and structures of the muscle microenvironment. In this review, we will discuss the normal role of these components as well as their role in coordinating the effective repair of skeletal muscle and their influence on muscle satellite cells.

### Skeletal Muscle Repair, Regeneration, and Adaptation

Adult skeletal muscle is frequently exposed to various stimuli, which has the potential to place the tissue into states of metabolic stress and elicit structural damage. Muscle damage can occur due to a variety of reasons, such as blunt force trauma, myotoxic agents, ischemia, dystrophy, eccentric exercise, or unaccustomed exercise. Despite skeletal muscle being susceptible to physical trauma and stress, it has a substantial capacity for regeneration due to the presence of satellite cells. Following the initial injury, there is degeneration of the muscle fibre through the breakdown of myofibrils and its sarcolemma [26]. Satellite cells activate by enlarging their nucleus, increasing DNA synthesis and the density of cytoplasmic organelles [26]. The next stage is the clearing of cell debris and damaged muscle fibres by neutrophils and pro-inflammatory macrophages [26]. The recruitment of these monocytes depends on local circulation and the capillary networks within skeletal muscle [26]. Lastly, satellite cells differentiate and fuse with existing myofibres donating their nuclei or can fuse with each other forming nascent myotubes [26]. Although satellite cells are the primary drivers of myogenesis within skeletal muscle, the efficient repair of muscle tissue relies on the complex interaction of multiple cell types. This review will focus on the supporting structures and cells that enable satellite cells to efficiently repair and regenerate skeletal muscle.

## 2. Muscle Stem Cells and Myogenesis

In 1961, Alexander Mauro discovered a population of cells that exist at the periphery of myofibres and appropriately called them satellite cells [1]. In homeostatic conditions, satellite cells are in a state of quiescence that is initiated and maintained by the continuous transcriptional activity of paired box protein 7 (Pax7) [1,27,28,29]. Once activated through stress or injury, satellite cells proliferate and either enter the early differentiation phase or return to quiescence to replenish the basal satellite cell pool [1,30]. The myoblast and myocyte progenies ultimately fuse with existing muscle fibres to donate their nuclei or fuse together to form de novo myofibres [1,30]. This process is known as the myogenic program and is regulated by a network of transcription factors collectively referred to as myogenic regulatory factors.

### 2.1. The Myogenic Program

The myogenic program is highly involved in the response to muscle damage. Myogenic regulatory factors are a group of transcription factors that, in conjunction with Pax7, regulate the action of satellite cells and their progeny, the myoblasts [28]. Myogenic differentiation 1 (MyoD), myogenic factor 5 (Myf5), myogenin (MyoG) and myogenic regulatory factor 4 (MRF4) direct satellite cell activation, proliferation, and differentiation [28]. The presence or absence of Pax7 and certain myogenic regulatory factors indicates which stage of the myogenic program the cells are in [28,31]. Pax7+/MyoD− cells are quiescent, Pax7+/MyoD+ cells are activated, and Pax7−/MyoD+ cells are committed to myogenic differentiation [27,28,29]. Once committed, the cells (now called myoblasts) start to express MRF4 and myogenin to initiate cell fusion [27,28,29]. Self-renewing satellite cells stop expressing myogenic regulatory factors and return to the quiescent state by upregulating Pax7 [27,28,29].

### 2.2. The Role of Satellite Cells in Regeneration and Repair

Satellite cells are necessary and sufficient to fully regenerate muscle following injury [3,4]. Seminal work in the field demonstrated that skeletal muscle has a high capacity for regeneration [2,32,33]. When minced muscle was transplanted into injured muscle, it regenerated new muscle fibres that were structurally and functionally sound [2,32,33]. We now know that this ability to repair was due to the presence and function of satellite cells. When satellite cells were genetically deleted from skeletal muscle prior to injury, the regenerative response was impaired [34]. However, when satellite cells were transplanted back into the injured muscle, regeneration was rescued [35]. One study found that even as few as seven transplanted satellite cells can completely regenerate a muscle ablated of host satellite cells via radiation [3]. These findings strongly suggest that satellite cells are essential to the process of muscle repair.

### 2.3. Plasticity of Satellite Cells

While satellite cells are largely considered unipotent, there is some evidence that they can be induced to differentiate into non-myogenic lineages in vitro [36,37]. Satellite cells isolated from mice can differentiate into osteocytes, adipocytes, and fibroblasts with the appropriate environmental cues; however, this is largely believed to be an in vitro phenomenon [36,37]. C2C12s mouse myoblasts exposed to bone morphogenetic protein-2 (BMP-2) transdifferentiated into osteoblastic progenitor cells, expressing markers, such as alkaline phosphatase (ALP) and osteocalcin, while inhibiting the formation of multinucleated myotubes [37,38,39]. Additionally, C2C12s will suppress MyoD and myogenin and undergo terminal osteogenic differentiation when exposed to beta-glycerophosphate [38].

Similarly, when cultured in an adipogenic cocktail, such as proliferative activated receptor gamma (PPARG) agonist or rosiglitazone, satellite cells transition into adipocytes [37,40]. Adding rosiglitazone, an insulin sensitizer and peroxisome PPARG agonist, activated PPARG expression and enhanced adipogenic differentiation of satellite cells [40,41]. Furthermore, satellite cells also possess the potential to transdifferentiate into fibroblasts. Satellite cells isolated from aged mice readily adopt a fibrogenic fate during proliferation through the activation of the *wnt* signalling pathway [42,43]. This suggests that the environment within aged muscle promotes a myogenic-to-fibrogenic conversion [42,43]. While it has been demonstrated that satellite cells can be considered multipotent in vitro, under normal conditions, they remain, by definition, a unipotent skeletal muscle stem cell [37,38,42,43,44].

## 3. The Supporting Cells of Skeletal Muscle Regeneration

Skeletal muscle is a complex tissue that contains myofibres and satellite cells but also multiple other cell types and structures, including resident immune cells, vasculature-associated cells, fibrogenic cells, and bone-marrow-derived cells. Although satellite cells can effectively regenerate muscle following injury, there are many cell types that influence and orchestrate the progression of satellite cells through the myogenic program. Although not necessary for regeneration, these cells can influence the myogenic program by releasing cytokines and growth factors that act on satellite cells, Figure 1. The supporting cells of the muscle microenvironment can influence satellite cells by releasing cytokines and growth factors and are considered essential for the effective repair of skeletal muscle damage.

### 3.1. Immune Cells

Muscle injuries trigger an inflammatory response that is essential for the proper repair of the tissue and includes the phagocytosis and degradation of damaged cells and pro-regenerative intercellular signalling. The predominant immune cells involved in muscle repair are neutrophils, macrophages, T cells, and mast cells, Figure 2 [24,25]. Following the initial damaging stimulus, resident leukocytes, including mast cells and macrophages, secrete cytokines and chemokines that quickly recruit neutrophils to the site of injury [45]. Neutrophils, in turn, release chemoattractants that recruit circulating immune cells and stimulate a pro-inflammatory environment to initiate the repair process [45]. One of these recruited cell types is circulating monocytes that differentiate into macrophages within skeletal muscle and are responsible for the phagocytosis of cellular debris [46,47,48]. There are three types of macrophages involved in the repair of skeletal muscle: tissue-resident macrophages, pro-inflammatory, and anti-inflammatory macrophages [46,47]. Each of these macrophage subpopulations serve distinct roles in the repair of muscle damage. More recently, regulatory T cells have been observed to infiltrate skeletal muscle following injury and were required for the restoration of muscle homeostasis [49,50]. For additional information on the immune system, we invite readers to consult Parkin and Cohen, 2001; Yatim and Lakkis, 2015; Tidball et al., 2010; Geissmann et al., 2011 [51,52,53,54].

#### 3.1.1. Neutrophils

Neutrophils have phagocytotic abilities and release chemokines and chemoattractants that recruit immune cells, from circulation to the site of injury [45]. Neutrophils are the first infiltrating immune cell to respond to muscle damage [55]. Injured skeletal muscle releases interleukin-6 (IL-6) and IL-8 to act as neutrophil chemotactic factors (i.e., chemokines that attract inflammatory cells) [55]. Neutrophils migrate to the site of injury by interacting with endothelial cells to transmigrate into the muscle [56]. This migration is dependent on reactive oxygen species (ROS) production, where ROS suppression prevents neutrophil–endothelial interactions [56,57]. However, the role of neutrophils within skeletal muscle extends beyond recruiting other immune cells via chemoattractants [58,59]. These immune cells also phagocytose muscle debris and exacerbate membrane lysis via nitric oxide (NO)-dependent processes to further aid in the recruitment of immune cells [58,59].

#### 3.1.2. Mast Cells

Mast cells are a muscle-resident leukocyte that play an important role in skeletal muscle repair. These cells primarily act to recruit neutrophils and other immune cells through several secreted factors, including macrophage inflammatory protein-2 (MIP-2), tumour necrosis factor (TNF)-α, and tryptase [60,61,62,63,64,65]. Particularly, MIP-2 (the murine homologue of human IL-8) has been highly implicated in the recruitment of neutrophils [66]. Mast-cell-deficient mice have impaired neutrophil accumulation, suggesting that mast cells contribute to the initiation of the muscle repair process [66]. Thus, muscle-resident mast cells are highly involved in the initial inflammatory response by mediating neutrophil recruitment to the site of injury.

Mast cells can regulate myogenesis, primarily through intercellular signalling driven by pro-inflammatory cytokines [63,64,65]. As mast cells respond to injury in the first few hours, they secrete TNF-α at low concentrations that stimulate satellite cell proliferation [60,62,63,64,65]. Secreted tryptase influences myogenesis by activating the protease-activated receptor-2 (PAR-2) on satellite cells, which has been shown to stimulate myoblast proliferation in vitro [63,64,65]. The release of tryptase by mast cells also helps regulate fibrosis by increasing fibroblast cell proliferation and collagen synthesis through PAR-2 [63,67]. Despite limited in vivo analyses on the regulatory action of mast cells, these immune cells appear to have an important role in regulating satellite cell and fibroblast activity during muscle regeneration.

#### 3.1.3. Tissue-Resident Macrophages

Another abundant leukocyte found within the skeletal muscle microenvironment is macrophages. Tissue-resident macrophages regulate skeletal muscle homeostasis and self-renew without the recruitment of blood-derived monocytes [66,68,69,70,71]. These immune cells can be triggered by muscle damage, and after the initial injury response by neutrophils, macrophages become the primary cell type in the regenerating muscle tissue. [58,59]. In response to acute injuries, tissue-resident macrophages act quickly to limit excessive inflammation by phagocytosing cellular debris in order to attenuate chemoattractant signals for neutrophil recruitment [70]. The importance of resident macrophages was demonstrated when it was shown that their depletion led to an increase in the number of necrotic myofibres and a reduction in the number of regenerating fibres [71]. Importantly, tissue-resident macrophages also serve to attract circulating macrophages via CC chemokine receptor 2 (CCR2) expression [72].

CC chemokine ligand (CCL2) and its receptor CCR2 are important for monocyte and macrophage recruitment and chemotaxis [24,73,74]. If either CCL2 or CCR2 is knocked out, the result is macrophage depletion [24,73,74]. In CCL2^−/−^ null mice, the absence of macrophages results in impaired satellite cell activity and muscle regeneration, demonstrating the importance of macrophage activity in myogenesis [75]. Macrophage depletion has also been linked to satellite cell activity, as seen by a reduced number of activated satellite cells and a decreased fibre size following regeneration [58].

#### 3.1.4. Pro-Inflammatory Macrophages

The immune system is an important component of regulating satellite cell activity through the secretion of cytokines and growth factors. The most significant contributor to the regulation of satellite cells is macrophages. Both pro- and anti-inflammatory macrophages influence satellite cell activity in a time-dependent manner. Pro-inflammatory macrophages promote the early stages of myogenesis (i.e., activation/proliferation), while anti-inflammatory macrophages promote the later stages of myogenesis (i.e., differentiation) [46].

Pro-inflammatory macrophages, also referred to as M1 macrophages, infiltrate the muscle shortly after neutrophils and secrete pro-inflammatory cytokines, which are essential during the initial stages of inflammation [24,25,76,77]. These macrophages secrete several cytokines and growth factors, such as IL-6, IL-1β, TNF-α, hepatocyte growth factor (HGF), tumour growth factor (TGF)- β, fibroblast growth factor (FGF), and vascular endothelial growth factor (VEGF) [46,49,78,79,80,81,82,83]. Several of these cytokines have direct effects on satellite cells and the myogenic program. IL-6, in particular, is classified as a myokine as it can be produced and released by myofibres and satellite cells and is heavily implicated in myogenesis and repair [84,85]. IL-6 is released by pro-inflammatory macrophages to promote satellite cell proliferation but also to stimulate the myoblast production of IL-6 through the secretion of IL-1β [46,55,83,86,87]. IL-1β also maintains satellite cell proliferation by inhibiting pro-differentiation signalling [46,86,87]. The cytokine TNF-α stimulates myoblast differentiation at low concentrations and promotes satellite cell proliferation at high concentrations [46].

Growth factors secreted by pro-inflammatory macrophages also have an important role in influencing satellite cell activity. HGF and IGF-1 expression are upregulated in regenerating muscle, suggesting that these growth factors may have a role in myogenesis [88,89,90,91]. HGF is produced in the early stages of regeneration to promote satellite cell activation and proliferation in a dose-dependent manner [88,89,90]. Pro-inflammatory macrophages are a source of IGF-1, which promotes repair by stimulating satellite cell activation, proliferation, and differentiation [85,88,92,93]. CCR2^−/−^ mice have diminished IGF-1 levels and impaired muscle regeneration, suggesting that macrophages are important for muscle repair and may be acting through IGF-1 signalling pathways [74]. Other macrophage-secreted growth factors can affect myogenesis in vitro; TGF-β has been found to inhibit myoblast differentiation, while fibroblast growth factor (FGF) can stimulate satellite cell proliferation [88,92,94]. Thus, growth factors secreted by macrophages are important regulators of satellite cell activity.

Pro-inflammatory macrophages are also important regulators of several cellular structures within the muscle microenvironment, independent of satellite cells. Macrophage-depleted mice have smaller myofibres that accumulate intramuscular necrotic, adipose, and fibrotic cells in response to injury [25,73,79,95]. Macrophage depletion also impairs VEGF-mediated angiogenesis and reduces the capillary-to-fibre ratio [25,73,79,95]. Pro-inflammatory macrophages also play a role in regulating excessive fibrosis through the phagocytosis of fibroblasts. Depletion of macrophages can lead to the rapid accumulation of fibroblasts, which, ultimately, increases fibrosis and impaired regenerative capacity [25,95]. TGF-β1 secretion by macrophages reduces the apoptosis of FAPs and induces differentiation into fibroblasts to stimulate the fibrotic regeneration [96]. These findings suggest that macrophages are not only important regulators of satellite cell activity, as they also affect several other cell types within skeletal muscle.

#### 3.1.5. Anti-Inflammatory Macrophages

Following the phagocytosis of cellular debris, macrophages transition from a pro- to anti-inflammatory phenotype. Anti-inflammatory macrophages limit the inflammatory reaction to muscle damage and are responsible for triggering the latter half of the muscle repair process [24,25,97]. They do this by secreting the cytokines IL-4, IL-10, IL-13, and TGF-β [25,75,79]. Several of these cytokines, including IL-13 and IL-4, have been linked to myotube fusion; studies have reported that myofibres lacking IL-4 or IL-4a receptors were smaller and contained fewer myonuclei [46,84,98]. Deng et al. (2012) found that IL-10 released by macrophages promotes an anti-inflammatory phenotype and promotes the proliferative stage of myogenesis [75]. These findings suggest a role for anti-inflammatory macrophage-secreted proteins in mediating satellite cell activity; however, there is a need for more in vivo analyses of their activity during skeletal muscle damage.

#### 3.1.6. T Lymphocytes

T lymphocytes, also known as T cells, are the main adaptive immune cell involved in muscle regeneration and exert their effects via interactions with macrophages or the secretion of cytokines [24,25,99]. The infiltration of T cells, specifically regulatory T cells (T regs), is involved in the switch from pro- to anti-inflammatory macrophages in skeletal muscle [24]. Burzyn and colleagues (2013) found that the depletion of T regs was accompanied by disorganized sarcolemmal structure, prolonged inflammation, increased fibrosis, and a reduction in the number of regenerating fibres that was associated with inhibited macrophage polarization [49,50]. T-reg depletion was also associated with an inhibited macrophage polarization during muscle repair [49]. Furthermore, T regs accumulate at approximately the same time as anti-inflammatory macrophages and have been suggested to play a role in the switch from pro- to anti-inflammatory macrophages via IL-10 secretion by promoting the expression of CD163 (i.e., anti-inflammatory phenotype) in macrophages [75].

In addition to modulating macrophage activity, T regs also have a role in regulating myogenesis. As T regs are recruited to the injured muscle in the later stages of regeneration, they are critical in promoting multiple stages of the myogenic program, either through the secretion of cytokines or participating in the macrophage phenotype switch. The secretion of IL-10 and TGF-β by T regs helps to promote the expansion of satellite cells, whereas IL-4 release promotes myotube fusion [98,100,101]. T regs also express amphiregulin, an epithelial growth factor that enhances myogenic differentiation in vitro [49].

### 3.2. Angiogenesis

Capillary density and distribution are crucial for muscle oxygenation and the delivery of secreted factors to skeletal muscle and are, therefore, a crucial component of the muscle microenvironment [102,103]. Severe injury not only damages myofibres but also fragments capillary networks within the muscle [104,105]. Thus, the re-vascularization of the tissue by endothelial cells and pericytes is important in supporting tissue remodelling and regeneration. Similar to other components of regeneration, vascular cells influence various cell types, including satellite cells, Figure 3.

#### 3.2.1. Capillaries

Capillaries have been shown to play an important role in the regulation of satellite cells, which are dependent on the vessels to transport important signalling factors to and from the cells [102,106,107,108]. When endothelial and satellite cells were co-cultured, there was enhanced satellite cell proliferation owing to the secretion of several growth factors, including IGF-1, HGF, basic fibroblast growth factor (bFGF), platelet-derived growth factor (PDGF)-BB, and VEGF [102,107]. The close spatial relationship between satellite cells and capillaries strongly suggests that they likely play a role in supporting satellite cell activity.

The spatial relationship of satellite cells and endothelial cells is also important in the revascularization of skeletal muscle post-injury [104,107,109]. Work from our group found that individuals with the shortest distance between capillaries and satellite cells had greater satellite cell activation and expansion following damage [107]. Furthermore, it has been shown that satellite cell proximity to blood vessels is associated with satellite cell self-renewal capabilities, as endothelial cells secrete Notch ligand D114 to induce quiescence in satellite cells to promote self-renewal [110]. Satellite cells and capillaries influence each other through the secretion of growth factors and regulatory factors [107]. VEGF, the primary angiogenic factor in skeletal muscle, is released by multiple cell types following injury [46,73,78,102,111,112]. This is sensed by endothelial cells and stimulates proliferation and capillary sprouting during the first 4 days post-injury [113]. When stimulated, satellite cells can also produce VEGF to induce endothelial cell chemotaxis, survival, and proliferation while also acting as a chemotactic factor for monocyte recruitment [78]. Satellite cells express *VEGF*-*A*, a VEGF gene member, to promote the transmigration of endothelial cells in vitro; blocking *VEGF*-*A* reduces the proximity of satellite cells to capillaries, indicating an important mechanism in satellite–endothelial cell proximity [73,110].

The reciprocal relationship between endothelial cells and satellite cells is further demonstrated by the release of angiotensin II following muscle damage. Angiotensin II, a regulator of angiogenesis during skeletal muscle repair, is a pro-angiogenic factor released by myoblasts that increases myotube length and branching in vitro [114]. Angiotensin II is also a regulator of satellite cell dynamics [114,115]. Treatment of C2C12 myoblasts with angiotensin II upregulated myogenic regulatory factors and chemotactic capacity [114,115]. Furthermore, the angiotensin-II-mediated chemotaxis of satellite cells is regulated through the reorganization of the extracellular matrix, as demonstrated by the increased invasion of myoblasts into gelatine-coated plates via increased matrix metalloproteinases-2 (MMP) activity [114,115].

Endothelial cells also aid in muscle repair by promoting immune-cell-mediated angiogenesis. VEGF and CCL2 secretion by endothelial cells acts as chemotactic factors for immune cell attraction [116]. These immune cells release MMPs that degrade the basement membrane in capillaries to allow sprouting to occur [117]. In CCL2 knockout mice, macrophage depletion was accompanied with a decreased capillary-to-fibre ratio and delayed angiogenesis [73,117]. These findings indicate the importance of the immune–endothelial cell relationship during angiogenesis and muscle regeneration.

#### 3.2.2. Pericytes

Pericytes are mesenchymal progenitor cells that encircle capillaries to regulate the permeability and constriction of microvessels. These cells participate in angiogenesis and are recruited via PDGF-BB signalling [21,113,118]. When injected into injured or dystrophic mouse muscle, pericytes enhance angiogenesis due to the production of the angiogenic factors bFGF and VEGF [20,21,111,118,119]. Studies have also found that pericytes can inhibit the division of endothelial cells via TGF-β activation or promote angiogenesis by secreting VEGF [23,112]. During skeletal muscle repair, endothelial cell and pericyte crosstalk is critical for angiogenesis. Following damage, pericytes migrate into the interstitial space and contribute to ECM remodelling through the secretion of collagen-I and MMP2 [16,118,120]. MMP2 acts to degrade ECM components to help promote satellite cell migration, which is necessary to reach the site of injured myofibres [61,121,122,123]. Thus, pericytes directly affect angiogenesis and satellite cell migration in order to facilitate muscle regeneration.

Although satellite cells are the primary myogenic progenitors in skeletal muscle, there are several cell types that have myogenic potential in the muscle microenvironment. Pericytes represent one such cell as they can enter the satellite cell niche and express myogenic genes and ultimately fuse with myofibres [18,19,124,125,126]. Birbrair and colleagues (2013) described two subpopulations of pericytes: type 1 pericytes, which are fibrogenic, and type 2 pericytes, which can participate in myogenesis [118]. When pericytes were transplanted into severe combined immune deficiency (SCID) *mdx* mice, they were able to fuse with myofibres and donate their nuclei [19,20,118,124]. This capability of type 2 pericytes to participate in myogenesis is reliant on environmental signalling; however, they are thought to have alternate myogenic differentiation kinetics, as compared to satellite cells [19,20,118].

The myogenic differentiation of pericytes is reliant on NF-κB, an inflammation-related transcription factor, that activates pericytes following injury [116]. In vitro experiments demonstrate that NF-κB activity in pericytes regulates myogenesis; when pericyte NF-κB activity is inhibited, there is increased pericyte fusion to myotubes [116]. Although pericytes have myogenic potential, the fusion of pericytes to muscle fibres is exceedingly rare under normal physiological conditions [19,20,118,124]. It appears that pericytes mainly influence regeneration by increasing angiogenesis and promoting satellite cell migration via ECM remodelling.

### 3.3. Fibrogenic Cells

The ECM is an essential component of skeletal muscle, providing support to several intramuscular structures, including myofibres, capillaries, and nerves. Fibroblasts and FAPs are the main cells involved in propagating fibrosis in skeletal muscle as they secrete ECM components for ECM remodelling. Following injury, skeletal muscle ECM undergoes extensive remodelling where fibrosis is a hallmark of regeneration [8,118,127]. However, extensive fibrosis caused by extreme damage, such as that observed in Duchenne muscular dystrophy, is associated with longer recovery times for muscle repair [8,118,127]. Thus, proper regulation of fibroblasts is essential for muscle repair.

#### 3.3.1. Fibroblasts

Within skeletal muscle, fibroblasts are the primary cells that maintain the ECM. They do so by secreting proteins such as collagen-I, which is an ECM component, and fibronectin that acts as a chemotactic factor for fibroblasts and macrophages [6,83,127]. Interestingly, fibroblasts also play a pivotal role in regulating satellite cell dynamics. After injury, fibroblasts have been found to expand rapidly in regions of regenerating myofibres in close proximity to satellite cells [8,127]. The ablation of fibroblasts results in premature satellite cell activation and differentiation, leading to a depletion of the satellite cell pool, suggesting an important role for fibroblasts in regulating myogenesis [127]. Furthermore, fusion is elevated when myoblasts are co-cultured with fibroblasts in transwell inserts, confirming that fibroblasts use intercellular signals to influence satellite cell activity [127]. Specifically, NF-κB release from fibroblasts upregulates iNOS expression within fibroblasts that promotes both myoblast proliferation and fusion [116,128].

Satellite cells also have the capacity to affect fibroblast activity. Satellite cell ablation in Pax7 knockout mice caused dysregulation of fibroblasts and increased connective tissue deposition [127]. Micro-RNAs, specifically miR206, are secreted by satellite cells to suppress excessive ECM deposition by downregulating the fibroblast expression of *Rrbp1*, a master regulator of collagen biosynthesis [129]. Furthermore, a study by Fry and colleagues (2017) demonstrated *Rrbp1* knockdown leads to decreased collagen biosynthesis, further illustrating the importance of satellite cells in regulating fibroblast activity [129].

#### 3.3.2. Fibro-Adipogenic Progenitors

FAPs are multipotent interstitial cells that can differentiate into fibroblasts or adipocytes based on the stimulus [11,15,44,130]. These cells can have either a positive or negative effect on muscle homeostasis based on their fate. For instance, the accumulation of adipocytes and fibrosis is common in muscular dystrophies, which hinders muscle regeneration [9,17]. FAPs play an integral role in regeneration by controlling fat deposition and fibrosis following injury, Figure 4.

FAPs are often observed in close proximity to damaged and regenerating myofibres, suggesting they may have a role in modulating muscle repair [14]. Studies have found that a reduction in FAP cell content is accompanied with impaired satellite cell expansion and impaired muscle regeneration [131,132]. However, excessive expansion of FAPs also results in impaired muscle regeneration; therefore, FAP activity must be tightly regulated to maintain skeletal muscle homeostasis [131]. FAPs are also capable of stimulating the differentiation of satellite cells in vitro; their co-culture with myoblasts led to an increase in terminally differentiated myoblasts [14]. These mesenchymal progenitors exert their influence on the myogenic program through the secretion of multiple factors, such as IL-6 and follistatin [14,96,133,134]. In particular, IL-6 levels are increased in FAPs following muscle damage to help promote satellite cell proliferation [14,134,135].

FAPs also have the potential to mediate myocyte fusion. Mozzetta and colleagues (2013) transplanted FAPs from young muscles into old muscles, which restored histone deacetylase inhibitor (HDACi) function and subsequently promoted differentiation and fusion of myoblasts [15,135]. The exposure of C2C12 to HDACi resulted in an increase in follistatin secretion by FAPs [15,136]. Follistatin acts by inhibiting myostatin, a negative regulator of skeletal muscle mass, activity to promote muscle growth [136,137,138]. Upregulation of follistatin within satellite cells is associated with enhanced regeneration, greater fusion index, and larger myotubes with more nuclei [136]. Follistatin also mediates the interaction between FAPs and satellite cells in *mdx* mice and is known to promote satellite cell fusion [15,134,136]. Adipogenic differentiation of FAPs can be inhibited by satellite cells [16]. Satellite cells can induce a fibrotic fate in FAPs through the secretion of TGF-β, a potent profibrotic cytokine [9,16]. Culturing FAPs with TGF-β stimulates their differentiation into fibroblasts [9,16]. Thus, satellite cells and fibrotic cells have a reciprocal relationship that promotes an efficient repair process in skeletal muscle.

Similar to fibroblasts, FAPs also have the potential to regulate the immune response. FAPs secrete several factors, including IGF-1, *Cxcl5*, *Cxc11*, and *Ccl7,* that recruit pro-inflammatory macrophages and neutrophils to help in the clearance of excessive fibroblasts during muscle repair [24]. FAPs are also a major source of IL-33 that act to regulate T-reg activity [50,68]. While there is still very limited research into the relationship between FAPs and immune cells, there appears to be some crosstalk between these cell types that has implications for muscle regeneration.

### 3.4. Bone-Marrow-Derived Cells

Bone-marrow-derived cells (BMDCs) are multipotent progenitors that have been found to contribute to multiple tissues, including skeletal muscle [139,140,141,142]. BMDCs have the capacity to differentiate into endothelial cells and can stimulate satellite cell differentiation via cell-to-cell contact [143,144,145]. BMDCs are considered “atypical” myogenic progenitors because they can contribute nuclei to myofibres, but not under normal, physiological conditions [141,146]. For instance, when transplanted into irradiated muscles, BMDCs were able to fuse with myofibres [141,146]. The contribution of BMDCs to myofibres begins by expressing myogenic regulatory factors and then progressing through the myogenic program [141,147,148]. While BMDCs can contribute to skeletal muscle by donating nuclei in dystrophic *mdx* mice, under normal, healthy conditions, there is no incorporation of BMDCs into skeletal muscle [142,146]. It appears that despite BMDCs having the potential to fuse with myofibres, they typically do not under normal, physiological conditions [141,144,146,149,150].

## 4. Conclusions

Satellite cells are indispensable for skeletal muscle regeneration; however, the interplay between satellite cells and other cells within the muscle microenvironment is essential for the effective and efficient repair of the tissue following damage. Myogenesis, inflammation, angiogenesis, and fibrosis are dependent on distinct cell types within skeletal muscle; however, these cells often communicate with each other to regulate these complex processes. The relationships between these cells are intricate, but it is evident that the efficient repair and regeneration of skeletal muscle is dependent on the continuous crosstalk between these cell populations. While we now know that regeneration is delicately orchestrated by several cell populations within skeletal muscle, these cells require further investigation, especially within human skeletal muscle.

## Figures and Tables

**Figure 1 cells-12-01968-f001:**
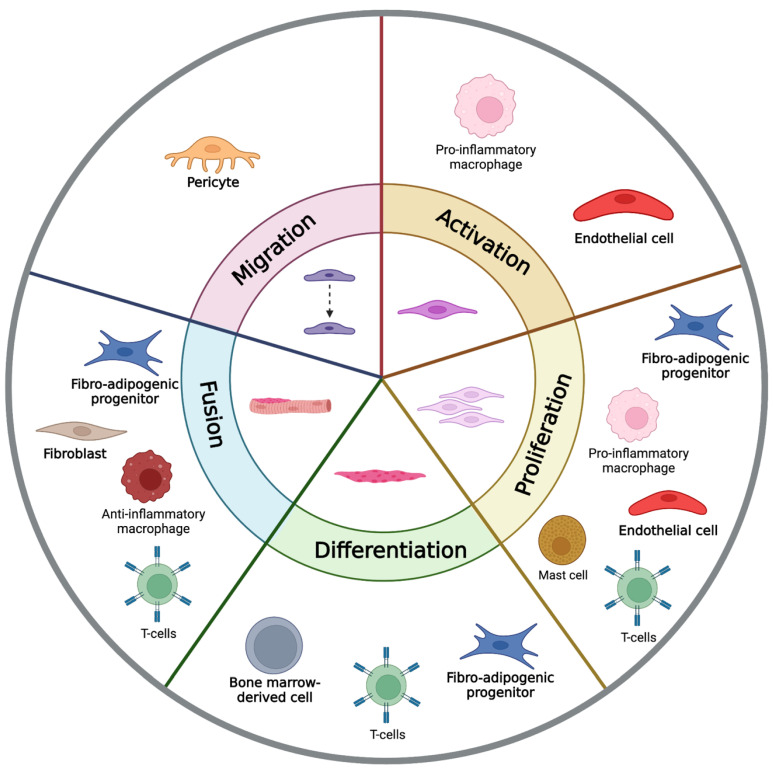
Supporting cell influences on the myogenic program. The different stages of satellite cell activity are all influenced by the supporting cells within skeletal muscle. Activation is promoted by pro-inflammatory macrophages and endothelial cells. Satellite cell proliferation is promoted by pro-inflammatory macrophages, endothelial cells, mast cells, T-cells, and fibro-adipogenic progenitors (FAPs). Differentiation is promoted by FAPs, T cells, and bone-marrow-derived cells. Fusion is promoted by FAPs, fibroblasts, anti-inflammatory macrophages, and T cells. Migration of satellite cells is promoted by pericyte and fibroblast activity.

**Figure 2 cells-12-01968-f002:**
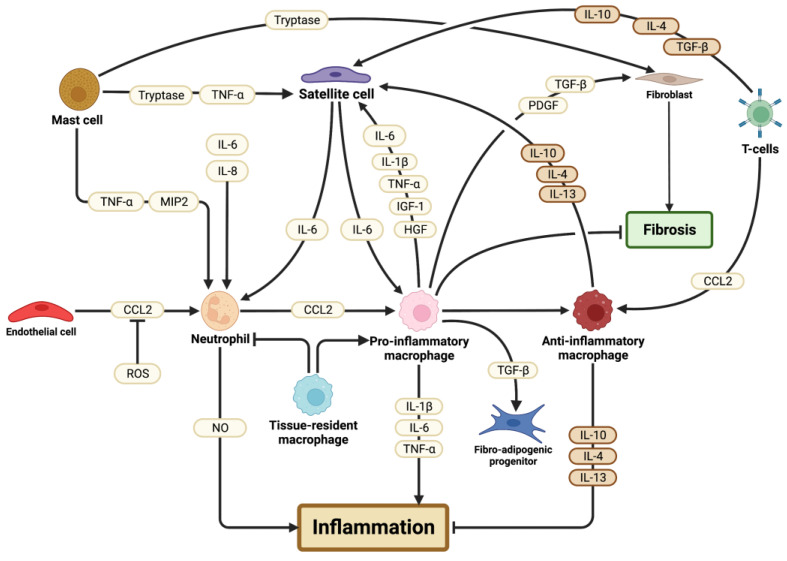
Schematic representation of the influences of immune cells in skeletal muscle regeneration. Inflammation is regulated by the immune cells: tissue-resident macrophages, neutrophils, pro- and anti-inflammatory macrophages, mast cells, and T cells. Supporting cells within the skeletal muscle microenvironment, such as satellite cells, fibro-adipogenic progenitors (FAPs), endothelial cells, and fibroblasts, are influenced by the immune cells. Secreted factors that influence cells are indicated in light yellow along the arrows and anti-inflammatory secreted factors are indicated in dark orange.

**Figure 3 cells-12-01968-f003:**
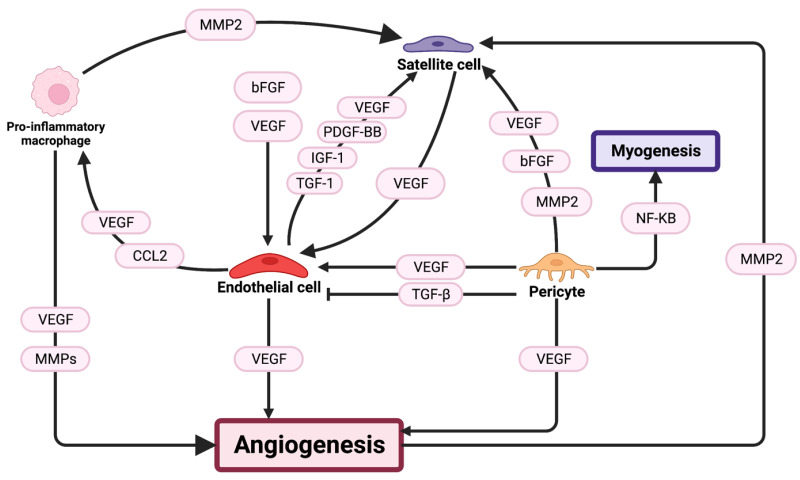
Schematic representation of the vasculature influences in skeletal muscle regeneration. Angiogenesis is controlled by endothelial cells and pericytes that are regulated by VEGF and PDGF-BB. Satellite cells and pro-inflammatory macrophages also act to regulate angiogenesis and are influences by interactions with vasculature-associated cells. Secreted factors that influence cells are indicated in pink along the arrows.

**Figure 4 cells-12-01968-f004:**
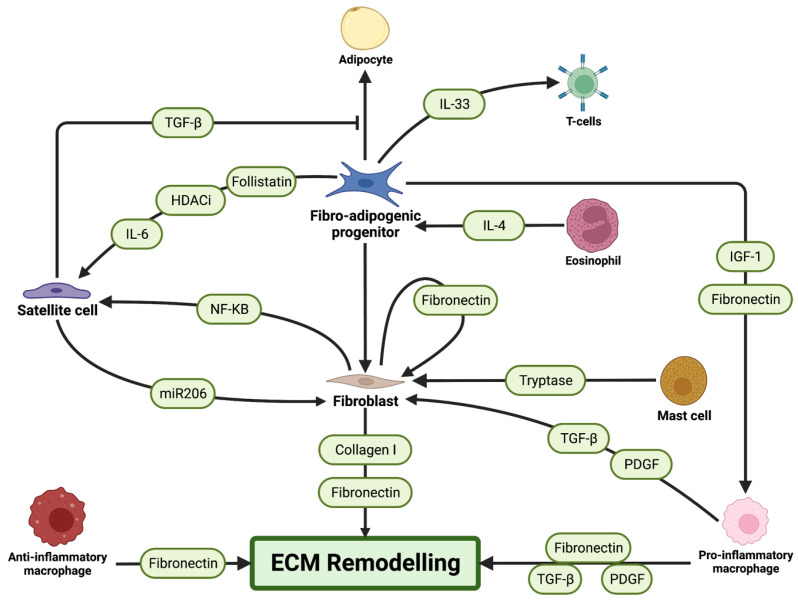
Schematic representation of the influences of fibrotic cells on skeletal muscle regeneration. Extracellular matrix (ECM) remodelling and fibrosis are mainly controlled by the fibrotic cells: fibroblasts and fibro-adipogenic progenitors (FAPs) that secrete ECM proteins to contribute to the ECM. Immune cells, such as macrophages, mast cells, and eosinophils, as well as satellite cells also influence fibrogenic cell activity and ECM remodelling. Secreted factors that influence cells are indicated in green along the arrows.

## Data Availability

The data presented in this review is from publicly available academic sources, which are cited in the reference list.

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
