# Peer review of "The Role of Supporting Cell Populations in Satellite Cell Mediated Muscle Repair"

_cells, 2023, doi:10.3390/cells12151968_

Round 1
Reviewer 1 Report
The authors are giving here an overview of cell-cell and cell-matrix interactions involved in skeletal muscle repair by satellite cells. The article mainly summarizes mouse data and displays limited novelty. Although the authors work in the human system, there are unfortunately no references from muscle organoids or at least hiPSC-derived cell cultures. This review could trigger assembloids using hiPSC-dreived myoblasts along with some of the other interacting cell types mentioned. The article is generally well written, but some phrases or expressions are not ideally suited for a research article, e.g. "Skeletal muscle has an incredible capacity..." (Abstract) should be rephrased avoiding exaggerations. Furthermore, "Skeletal muscle is a highly plastic organ..." (Introduction) sounds inappropriate, although I totally agree regarding its plasticity. Finally, the legend of Figure 2 refers to yellow and orange colors. However, in my version, these appear as beige and brown (?).
The article is generally well written, but some phrases or expressions are not ideally suited for a research article, e.g. "Skeletal muscle has an incredible capacity..." (Abstract) should be rephrased avoiding exaggerations. Furthermore, "Skeletal muscle is a highly plastic organ..." (Introduction) sounds inappropriate, although I totally agree regarding its plasticity.
Reviewer 2 Report
This paper reviews the role of supporting cells for muscle repair. I think this paper is clearly written and organized. I would like to make one correction. Please correct italic to non-italic of “2. Muscle stem cells” and reconsider the tittle of section 2 because I think the tittle mismatches the contents of 2.2 and 2.3. I don’t think any other major modifications are needed.
Reviewer 3 Report
The review is very well written. The aim of the review is an important topic for researchers in the muscular field. The authors reflect on the normal role of supporting cell populations, their role in coordinating effective skeletal muscle repair and their influence on muscle satellite cells. The structure is logical and very easy to read. Overall, a very balanced description of the processes in the muscle cell without going into too much detail. The text is never too superficial, but gives the interested reader a very good overview of the interrelationships of muscle cell repair. The illustrations are very easy to understand and well labelled. They fit very well into the overall concept of the text.
Experts in the field would like to see more in-depth information with citations of the latest literature in each chapter, as there has been a lot of new literature on this subject in the last two years.
